# Photosynthetic Gains in Super-Nodulating Mutants of *Medicago truncatula* under Elevated Atmospheric CO_2_ Conditions

**DOI:** 10.3390/plants12030441

**Published:** 2023-01-18

**Authors:** Rose Y. Zhang, Baxter Massey, Ulrike Mathesius, Victoria C. Clarke

**Affiliations:** 1Research School of Biology, Australian National University, Canberra, ACT 2601, Australia; 2Tasmanian Institute of Agriculture, University of Tasmania, Sandy Bay, TAS 7005, Australia

**Keywords:** symbiosis, nitrogen, photosynthesis, elevated CO_2_, super-nodulating mutants, *Medicago truncatula*

## Abstract

Legumes are generally considered to be more responsive to elevated CO_2_ (eCO_2_) conditions due to the benefits provided by symbiotic nitrogen fixation. In response to high carbohydrate demand from nodules, legumes display autoregulation of nodulation (AON) to restrict nodules to the minimum number necessary to sustain nitrogen supply under current photosynthetic levels. AON mutants super-nodulate and typically grow smaller than wild-type plants under ambient CO_2_. Here, we show that AON super-nodulating mutants have substantially higher biomass under eCO_2_ conditions, which is sustained through increased photosynthetic investment. We examined photosynthetic and physiological traits across super-nodulating *rdn1-1* (Root Determined Nodulation) and *sunn4* (Super Numeric Nodules) and non-nodulating *nfp1* (Nod Factor Perception) *Medicago truncatula* mutants. Under eCO_2_ conditions, super-nodulating plants exhibited increased rates of carboxylation (V_cmax_) and electron transport (J) relative to wild-type and non-nodulating counterparts. The substantially higher rate of CO_2_ assimilation in eCO_2_-grown *sunn4* super-nodulating plants was sustained through increased production of key photosynthetic enzymes, including Rieske FeS. We hypothesize that AON mutants are carbon-limited and can perform better at eCO_2_ through improved photosynthesis. Nodulating legumes, especially those with higher nitrogen fixation capability, are likely to out-perform non-nodulating plants under future CO_2_ conditions and will be important tools for understanding carbon and nitrogen partitioning under eCO_2_ conditions and future crop improvements.

## 1. Introduction

As atmospheric carbon dioxide (CO_2_) levels continue to rise [1], plant productivity and crop yields will be impacted. As the primary photosynthetic substrate, elevated CO_2_ (eCO_2_) levels are anticipated to alter photosynthetic efficiency, as well as having potentially broader influences on developmental and physiological processes. The eCO_2_ conditions reduce the frequency of photorespiration, enhancing photosynthetic rates and reducing respiratory costs [2]. This enhanced photosynthesis translates into positive growth responses for C_3_ plants, with an average 20% increase in above-ground dry matter production under higher CO_2_ concentrations [3]. Accordingly, C_3_ crop species exhibit a 17% increase in yield under eCO_2_, with cotton yields even increasing by over 40% [3]. While such yield responses may seem promising, high CO_2_ concentrations produce grains with lower protein and amino acid content when compared to ambient-grown grains [4]. If crop plants are unable to maintain optimal carbon:nitrogen (C:N) ratios under eCO_2_, the cost of reduced nutritional quality will outweigh the benefits of increased grain yield.

Long-term eCO_2_ conditions tend to induce a characteristic photosynthetic acclimation response, where the overall benefit of eCO_2_ becomes reduced due to the downregulation of photosynthesis. This acclimation effect is most commonly observed through declines in the maximum carboxylation rate of Rubisco (V_cmax_), as well as the maximum electron transport rate contributing to RuBP regeneration (J) [5]. There are two major hypotheses surrounding this acclimation or downregulation response [6], both of which are associated with the sensitivity of Rubisco to CO_2_ and nitrogen (N) availability [7]. The “sink limitation” hypothesis suggests that greater carbon assimilation under eCO_2_ increases carbohydrate concentration in leaves, resulting in the downregulation of photosynthetic enzyme synthesis. On the other hand, the “nitrogen limitation” hypothesis instead postulates that photosynthetic acclimation arises from nutrient, specifically nitrogen, limitation under greater carbon assimilation. Indeed, C_3_ plants demonstrate significantly lower tissue %N when grown with eCO_2_ compared with ambient CO_2_ (aCO_2_) conditions [8].

Studies indicate that in comparison with non-legumes, legumes experience greater stimulation of photosynthesis under eCO_2_ [3,9]. The key factor contributing to this response is the development of symbiotic nitrogen fixation (SNF) in legumes. SNF allows legumes to obtain biologically available forms of nitrogen from nitrogen-fixing bacteria known as rhizobia. During the initiation of SNF, legumes secrete rhizobia-specific flavonoids from their roots. These compounds activate the expression of rhizobial *Nod* genes, followed by Nod factor perception by the host legume, bacterial infection, and nodule development [10]. Once rhizobia successfully infect the host legume, they differentiate into bacteroids and inhabit a specialized root structure known as the nodule. In exchange for bio-available nitrogen compounds, the host legume provides the rhizobia with carbohydrates in the form of malate and other organic acids [11].

Not only does SNF provide legumes with a greater pool of available nitrogen, it also acts as a sink for carbohydrates assimilated in the leaves. Under eCO_2_ conditions, SNF compensates for cellular reductions in N while preventing the downregulation of key photosynthetic enzymes including Rubisco by acting as a carbon sink. Substantial experimental evidence supports the enhanced performance of nodulating legumes under eCO_2_ [12,13]. Some studies report greater nodule numbers and nodule biomass under eCO_2_ [7], while others observe far higher levels of total leaf N, specific leaf area (SLA), and harvestable biomass [14,15].

The responsiveness and sensitivity of nodulation to eCO_2_ suggests that photosynthesis and nodulation are closely regulated and interdependent. At a fundamental level, photosynthetic machinery imposes a heavy nitrogen cost upon plants, with chloroplasts composing 75% of average C_3_ leaf nitrogen [16]. Rubisco alone makes up 40% of soluble protein and 20% of leaf nitrogen in C_3_ leaves [16]. Likewise, nodulation imposes a heavy carbon cost upon plants, with the maintenance of active nodules estimated to require at least 6 g of carbon per gram of nitrogen obtained, although this depends on the environmental conditions and efficiency of the SNF [17,18]. To control the high carbohydrate demand of nodules, legumes display autoregulation of nodulation (AON) to restrict nodules to the minimum number needed to sustain nitrogen supply under current photosynthetic levels [19].

AON has been hypothesized to act as a systemic signaling mechanism, requiring a complex signal transduction process between the roots and the shoots [20,21]. In the model legume *Medicago truncatula*, the initial infection of rhizobia into the roots induces the local production of CLE12 and CLE13 peptides [22,23]. The CLE12 peptide is activated by tri-arabinosylation in the root by the enzyme RDN (Root Determined Nodulation) [20,24]. Both peptides are then transported into the shoot via the xylem, where they are perceived by the leucine-rich repeat receptor kinase (LRR-RK) SUNN (Super Numeric Nodules) [25]. Disruption to various components of this signaling mechanism can result in abnormal nodule development. Super-nodulating legumes have a deficient AON pathway and are unable to regulate their nodule numbers [19]. Instead, they produce excessive numbers of nodules at high physiological cost to the plant. At ambient CO_2_, the *M. truncatula* AON mutants *sunn4* and *rdn1-1* are characterized by numerous small nodules, lower biomass but higher %N in shoot and root tissues, at least at low external nitrogen supply [26], similar to AON mutants in other species [27]. On the other hand, non-nodulating legumes are defective in the processes required to initiate rhizobial symbiosis, preventing them from exhibiting SNF.

In this study, we assessed various phenotypic traits of the model legume *Medicago truncatula* across wild-type (cultivar A17), super-nodulating mutants *rdn1-1* and *sunn4*, and the non-nodulating mutant *nfp1* [28] under two different CO_2_ treatments: ambient (500 µmol mol^−1^ CO_2_) and elevated (1000 µmol mol^−1^ CO_2_). We assessed CO_2_ assimilation rate, chlorophyll fluorescence and plant physiology parameters alongside levels of the photosynthetic enzymes to elucidate the extent of nitrogen allocation toward photosynthetic machinery in AON mutants.

## 2. Results

### 2.1. Super-Nodulating Plants Are More Photosynthetically Responsive to Elevated CO_2_

CO_2_ response curves revealed similar rates of CO_2_ assimilation in ambient-grown (aCO_2_) *rdn1-1* and *sunn4* super-nodulating mutants relative to ambient-grown wild-type plants (Figure 1A). In contrast, *rdn1-1* and *sunn4* mutants exhibited significantly higher CO_2_ assimilation rates than wild-type plants when grown under elevated CO_2_ conditions (eCO_2_), while non-nodulating *nfp1* plants exhibited the lowest CO_2_ assimilation rates across both CO_2_ treatments (Figure 1B). Both measured and model-derived estimates of maximum electron transport (J) support this trend, with eCO_2_ *rdn1-1* and *sunn4* mutants displaying significantly higher J estimates relative to eCO_2_ wild-type and *nfp1* plants (Figure 2B & Table 1). When grown under eCO_2_, *rdn1-1* and *sunn4* mutants also displayed increased maximum electron transport and maximum Rubisco activity (V_cmax_) (Figure 2A) when compared with their ambient-grown counterparts.

Western blot analysis was performed on a subset of samples grown at both CO_2_ concentrations. The *sunn4* super-nodulating mutant was not significantly different to wild-type in the amount of rubisco small subunit protein detected, under both CO_2_ conditions (Figure 3B). Total rubisco content (measured by CABP-binding assay) was higher in *sunn4* mutants than wild-type controls at both CO_2_ concentrations but was not significantly different between CO_2_ concentrations within any genotype (Figure 2C). The relative Rieske protein content, as determined by western blot, of eCO_2_ *sunn4* mutants was almost two-fold higher than in eCO_2_ wild-type plants, and three-fold higher than eCO_2_ *nfp1* mutants (Figure 3A). Across all nodulation genotypes, plants with higher Rubisco and Rieske content tend to exhibit correspondingly higher rates of maximum Rubisco activity (V_cmax_) and maximum electron transport (J), respectively (Figure 4A,B). Active Rubisco content also shows positive affiliation with increased leaf %N (Figure 4C).

In contrast with generally observed improvements in photosynthetic activity in super-nodulating mutants under eCO_2_, wild-type plants grown under eCO_2_ conditions displayed significantly higher concentrations of chlorophyll A than super-nodulating plants (Figure 2D). Stomatal conductance was largely unresponsive to CO_2_ concentration and plant genotype, with comparable values across all treatments (Table 1).

### 2.2. Super-Nodulating Plants have Increased Biomass and Leaf Nitrogen Content under Elevated CO_2_ Conditions

Wild-type, *rdn1-1,* and *sunn4* plants exhibited a significant three-to-four-fold increase in above-ground biomass (AGB) when grown under elevated CO_2_ conditions compared to ambient CO_2_ conditions (Figure 2E). Within the elevated CO_2_ treatment, *nfp1* mutants had significantly lower AGB than wild-type plants, while the AGB of *rdn1-1* and *sunn4* mutants was significantly higher (10–35% increase) than wild-type plants.

Under both CO_2_ conditions, *nfp1* plants exhibited higher leaf carbon:nitrogen (C:N) ratios than super-nodulating plants, with this difference being significant at eCO_2_ (Figure 2F).

Interestingly, the carbon content (%C) in leaves of wild-type plants experienced a significant decline when grown at eCO_2_ conditions, while neither *nfp1* nor super-nodulating plants demonstrating any shift in carbon content under eCO_2_ (Table 1). At eCO_2_, %N in leaves was also significantly increased in super-nodulating plants compared to the wild-type control (Table 1).

## 3. Discussion

This study assessed carbon and nitrogen assimilation in super-nodulating (*rdn1-1* and *sunn4*) and non-nodulating (*nfp1*) mutants of *Medicago truncatula* under ambient and elevated CO_2_ conditions to better understand the role of symbiotic nitrogen fixation (SNF) in mediating photosynthesis under eCO_2_ levels.

Wild-type plants, with the ability to autoregulate nodule number, were expected to increase their biomass production, N fixation, and N assimilation under high CO_2_ as a result of a greater carbon supply to the nodules [29]. We found that biomass and chlorophyll content were increased under eCO_2_ conditions (Figure 2, Table 1). We hypothesized that this response to elevated CO_2_ would be modulated by higher CO_2_ assimilation rate and potentially increased N allocation toward photosynthetic machinery, but while trending higher, these differences were not significant in wild-type plants between the two CO_2_ conditions (Table 1). Leaf N and carbon (C) content, however, were reduced in the leaves of wild-type plants grown under eCO_2_, with the C:N ratio trending higher but not significantly different (Table 1). This suggests that despite having increased N availability due to SNF, wild-type *Medicago* plants may still be N limited under eCO_2_ conditions and are not able to effectively re-direct additional C assimilated in photosynthesis to SNF. Leaf ^15^N content (a proxy for SNF) measurements were unreliable in this study so N fixation impacts could not be determined (data not shown).

The super-nodulating *rdn1-1* and *sunn4* mutants were expected to perform sub-optimally at ambient CO_2_ levels due to the high C cost of super-nodulation, though we observed no significant decrease in photosynthetic or physiological traits compared to wild-type plants at ambient CO_2_ (Table 1). This may be due to the ambient CO_2_ level in our growth chambers being at 500 µmol mol^−1^ CO_2_, enabling the super-nodulators to maximize SNF. Previous analyses of *sunn4* and *rdn1-1* mutants in *Medicago* also did not find significant decreases in shoot biomass at ambient (400 µmol mol^−1^ CO_2_) levels compared to the A17 wild-type plant, with *rdn1-1* even having a higher shoot biomass than wild-type plants at 550 µmol mol^−1^ CO_2_ [30]. At elevated CO_2_ levels, however, the *sunn4* and *rdn1-1* super-nodulating mutants had substantially increased photosynthetic activity and biomass accumulation. *Sunn4* and *rdn1-1* had significantly increased maximum carboxylation (V_cmax_) and electron transport rate (J) when grown at eCO_2_ compared to aCO_2_, which was not observed in wild-type plants (Figure 2A,B).

Across all nodulation genotypes, plants with higher Rubisco and Rieske content also had correspondingly higher rates of maximum Rubisco activity (V_cmax_) and maximum electron transport (J), respectively (Figure 4A,B). The substantially higher rate of CO_2_ assimilation in eCO_2_-grown *rdn1-1* and *sunn4* plants relative to wild-type plants is sustained by increased production of key photosynthetic enzymes, with the *sunn4* super-nodulating mutant having significantly increased Rieske protein levels compared to the wild-type control (Figure 3B). Rieske FeS (PetC) forms one of four major subunits contained within the Cytochrome *b_6_f* (Cyt*b_6_f*) monomer, and in its dimeric form, Cyt*b_6_f* forms one of the major protein complexes within the electron transport chain of C_3_ plants [31]. Increased Rieske expression has been associated with an upregulation of Cyt*b_6_f* as well as improved CO_2_ assimilation [32], making it a useful biochemical proxy for electron transport rate.

The *sunn4* super-nodulating mutant also had increased total Rubisco content at both aCO_2_ and eCO_2_ compared to wild-type controls (Figure 2E). Rubisco is composed of eight small and eight large subunits, and is the major photosynthetic enzyme, responsible for catalyzing the reaction of CO_2_ with ribulose 1,5-bisphosphate (RuBP) during the carbon assimilation process [33]. *Rdn1-1* samples were not able to be assessed for Rubisco or Rieske content, but it would be expected they would also have increased levels similar to *sunn4* due to their increased carboxylation and electron transport rates as determined through CO_2_ response curves (Figure 2A,B). It is important to note, however, that the wild-type control (A17) in this experiment has the same background as *rdn1-1* (Jemalong A17), while the *sunn4* line has a related, but different, background (Jemalong J5). While *rdn1* and *sunn4* had similar physiological responses to eCO_2_, and there are no reported physiological differences between the lines [25], it is possible that the biochemical differences between *sunn4* and WT A17 plants may be influenced by differences in their background genotype.

This upregulation of photosynthetic enzymes and CO_2_ assimilation is unique to the super-nodulating genotype and lies in contrast to widely reported acclimation responses to eCO_2_ across nodulating legumes that exhibit autoregulation of nodulation (AON). A meta-analysis by Ainsworth and Long [3] evaluated over 15 pairwise comparisons across three legume species grown at ambient and elevated CO_2_ levels, and found an average decline in V_cmax_ and J by ~6% and ~13%, respectively. Likewise, when grown under seven different CO_2_ concentrations, soybean exhibit dramatically lower V_cmax_ and J beyond optimal CO_2_ concentrations of 400–600 µmol mol^−1^ CO_2_ [34]. Calculations based on legumes grown under free-air CO_2_ enrichment (FACE) experiments also suggest a 10% reduction in leaf rubisco content between current and elevated CO_2_ concentrations [5]. As such, super-nodulating mutants of *M. truncatula* appear to be capable of overcoming eCO_2_-induced photosynthetic acclimation.

The non-nodulating *nfp1* mutant is entirely unable to form nodules and had similar physiology to non-legumes under high CO_2_, with substantial N limitation resulting in reduced photosynthetic performance and decreased biomass (Table 1). Non-nodulating legumes, such as *nfp1,* display higher C:N ratios under eCO_2_ due to N limitation [4,35]. In this study, while C:N ratios tended higher in the non-nodulating *nfp1* mutant, this difference was not significant, likely due to the reduced sample replication possible in these very small plants (Figure 2D, Table 1). The super-nodulating *rdn1-1* and *sunn4* mutants were observed to maintain relatively constant C:N ratios, however (Figure 2D, Table 1). This unique capacity for *rdn1-1* and *sunn4* mutants to enhance CO_2_ assimilation rate while maintaining C:N ratios arises from the greater N availability provided by super-nodulation [30]. Active rubisco content also shows positive affiliation with increased leaf %N across the lines (Figure 4C). With Rubisco accounting for 20% of leaf nitrogen content in C_3_ plants [16], this correlation provides further support that additional N obtained through SNF in the super-nodulating legumes is able to be invested in photosynthesis

In this study, the impact of greater N assimilation on photosynthetic enzyme content could not be conclusively established through analyzing the portion of N assimilated through SNF, as root δ^15^N values may have been contaminated from the presence of potting vermiculite (Stuart-Williams, personal communication). Where possible, the cultivation of legumes in hydroponic rather than soil medium may allow for more extensive and reliable ^15^N quantification, and this approach would also make nodules more readily accessible for morphological or phenotypic assays. ^13^C analysis of C assimilation integration data was also not possible in this study due to the differing ^13^C signature of the eCO_2_ air supplementation compared to ambient CO_2_.

The physiological outcome of enhanced C and N assimilation in super-nodulating legumes is evident in the significant above-ground biomass accumulation in *rdn1-1* and *sunn4* plants at eCO_2_ conditions (Figure 2C), which has also been observed in *rdn1-1* previously [30], but not in soybean super-nodulating mutants grown at high light and/or CO_2_ [36,37]. While Ainsworth, Davey [38] report greater shoot dry weight (SDW) improvement in non-nodulating soybeans (76% improvement) compared to nodulating soybeans (28% improvement), this study reports greater relative AGB increase in super-nodulating and wild-type *M. truncatula* plants compared to non-nodulating *nfp1* plants. This discrepancy may arise from a greater rhizobial carbon demand in soybeans compared to *M. truncatula* or suggests the potential for different C and N allocation regimes between legume species. Nodulating soybean plants may increase C allocation to seeds under eCO_2_, while nodulating *M. truncatula* plants may instead prioritise shoot growth. Indeed, soybean seeds produced under elevated CO_2_ conditions have been found to accumulate higher C and isoflavone content compared to seeds produced under ambient CO_2_ conditions [39,40].

Across both ambient and elevated CO_2_ conditions, *rdn1-1* and *sunn4* mutants performed comparably within all the phenotypic traits assessed in this study. This indistinguishability between *rdn1-1* and *sunn4* lies in contrast to results from Qiao, Miao [30], wherein eCO_2_-grown *sunn4* mutants exhibited significantly reduced shoot and root biomass, as well as total N per plant, fixed N per plant, and fixed N per nodule when compared with eCO_2_-grown *rdn1-1* mutants. Greater phenotypic differences between *rdn1-1* and *sunn4* would be expected given the distinct nodule responses that have been previously documented for these two super-nodulating mutants, with greater nodule mass and number in *sunn4* compared to *rdn1-1* mutants at ambient [26] and elevated CO_2_ [30]. Differences in experimental design may account for this disparity, as Qiao et al. examined phenotypic responses in soil and under a CO_2_ concentration of 300–850 µmol mol^−1^ CO_2_, while CO_2_ treatments in this study ranged from 500–1000 µmol mol^−1^ CO_2_. Physiological responses in plants often exhibit a bell-shaped curve with respect to environmental variables such as CO_2_ concentration [34], with comparisons between different points on the curve thus yielding different outcomes. In order to address this discrepancy, future studies could further elucidate subtle differences between *rdn1-1* and *sunn4* super-nodulating mutants through cultivation under a wider range of CO_2_ concentrations.

Ultimately, through comparisons between *rdn1-1/sunn4*, wild-type, and *nfp1* plants, this study provides compelling evidence for the capacity of symbiotic nitrogen fixation to support greater photosynthetic assimilation under elevated CO_2_ conditions. Under the elevated CO_2_ conditions in this study (1000 µmol mol^−1^ CO_2_), super-nodulating legumes were able to re-invest the additional N fixed through SNF back into photosynthesis to overcome C limitations. The ability of legumes to obtain a reliable source of N from SNF, and the ability of super-nodulating legumes to further enhance these benefits will inform efforts to providing reliable and nutritious crop yields under future elevated CO_2_ conditions. Future field-based experiments, such as those conducted using FACE techniques, will be critical in determining whether presently observed improvements to photosynthesis and plant growth can be sustained in the absence of controlled laboratory conditions.

## 4. Materials and Methods

### 4.1. Plant Material

*Medicago truncatula* seeds of the wild type line Jemalong A17, the non-nodulation mutant *nfp1* [derived from A17, 28], the *sunn4* mutant [derived from the related genotype Jemalong J5, 25], and the *rdn1-1* mutant [derived from Jemalong A17, 20] were used. The *rdn1-1* mutant has a 103-kb deletion that includes *RDN1* as well as 17 other annotated genes [20]. So far, no obvious phenotypic differences have been reported between the wild type cultivars Jemalong A17 and Jemalong J5 [25].

Seeds were chemically scarified in concentrated H_2_SO_4_ for 3 min, thoroughly washed with water, and surface-sterilized in NaClO for 5 min before being washed in sterile Milli-Q water. After germination at 25 °C overnight, seedlings were sown in pots filled with vermiculite. One seedling was grown in each pot. Pots were arranged in a random block design with ten replicates (ten blocks within each growth room). Vermiculite was used because it contains little to no residual nitrogen, allowing nitrogen concentration to be externally controlled. A volume of 50 mL of nutrient solution (Fåhraeus Medium; composition (mg/L) of KH_2_PO_4_, 217; K_2_SO_4_, 74; CaCl_2_·2H_2_O, 244; MgSO_4_·7H_2_O, 122; NaB_4_O_7_, 1.6; MnSO_4_·H_2_O, 6; ZnSO_4_·7H_2_O, 8; CuSO_4_·5H_2_O, 6; CoCl_2_, 0.4; Na_2_MoO_4_, 0.4; FeCl_3_, 0.6.) was provided to each pot every 3 days. Once a week, each pot also received 1 mL of 0.5 mM KNO_3_ that was enriched with ^15^N at 1% atom excess. This enriched the growth medium in ^15^N, enhancing the difference in ^15^N between soil-derived and atmosphere-derived nitrogen and allowing for estimation of nitrogen derived from atmospheric nitrogen [41].

Two weeks after germination, seedlings were inoculated with a cultured suspension of the *M. truncatula* symbiont *Sinorhizobium meliloti* strain 1022. The rhizobia were first grown on a plate of solid BBM agar [42] and then a single colony was picked and grown overnight in liquid BMM in a 50 mL Falcon tube. After 16 h, the optical density (OD) of the rhizobial suspension was measured with a spectrophotometer at 600 nm and adjusted to an OD_600_ of 0.1 to standardize rhizobial density. Each pot received 1 mL of this suspension.

### 4.2. Growth Conditions

From 2 weeks post-germination, plants were grown under controlled-environment conditions in two growth rooms (Controlled Environment Facility, ANU). The ambient CO_2_ (aCO_2_) growth chamber was maintained at ~500 µmol mol^−1^ CO_2_ (it was not possible to maintain the CO_2_ to a value closer to the current ambient CO_2_ of 414 µmol mol^−1^ CO_2_), while the elevated CO_2_ (eCO_2_) growth chamber was maintained at ~1000 µmol mol^−1^ CO_2_ (average concentration over 24 h). Both chambers had 16 h:8 h light:dark cycles, with day temperatures of 23 °C, ~250 µE light intensity, and a set humidity of 60% (actual humidity levels varied between 50–70%).

### 4.3. Photosynthetic Activity

CO_2_ response curves of carbon assimilation rate and chlorophyll fluorescence were measured together with a LI-6800 portable photosynthesis system (LI-COR Biosciences, Lincoln, NE, USA), at a leaf temperature of 25 °C, irradiance of 1500 µmol quanta m^−2^ s^−1^, relative humidity of 55%, 21% O_2_, and varying reference CO_2_ concentrations (0, 50, 75, 100, 200, 300, 400, 600, 800, and 1200 µmol mol^−1^). All measurements were taken from the youngest, fully expanded trifolate leaf on the longest shoot of 7-week-old plants. Each measured trifolate leaf was photographed alongside a reference ruler, and total leaf area was extracted using ImageJ Java 1.8.0_172 [43] to correct for leaf area within the LI-6800 chamber. Curves were analyzed to derive estimates of maximum Rubisco activity, V_cmax_, and the rate of electron transport, J [44]. Fluorescence-based measurements of electron transport rate (ETR) are also considered equivalent to J when divided by four (J = ETR/4). Leaf discs were harvested from measured plants, oven-dried, and weighed to determine specific leaf area (SLA), and above-ground biomass (AGB) was determined from mature plants harvested at 9 weeks of age.

### 4.4. Biochemical Assays

For biochemical analysis of leaves, leaf discs of 0.353 cm^2^ corresponding to the area where gas exchange was measured were collected and frozen immediately in liquid N_2_. Active Rubisco content was estimated from leaf discs by the irreversible binding of [^14^C]2-carboxy-D-arabinitol 1,5-bisphosphate to the fully carbamylated enzyme, as described by Ruuska et al. [45].

In order to isolate leaf proteins, protein extract derived from the active Rubisco assay was diluted into 4x SDS Sample buffer containing 0.25 M tris-HCl pH 6.8, 40% (*v/v*) glycerol, 8% SDS, 4% bromophenol blue, 0.5% betamercaptoethanol, and incubated at 95° C for 5 min. Samples were loaded on a leaf area basis and separated by polyacrylamide gel electrophoresis (Nu-PAGE 4–12% Bis-(2-hydroxyethyl)-amine-tris(hydroxymethyl)-methane (Bis-Tris) gel, Invitrogen, Life Technologies Corporation, Carlsbad, CA) in running buffer (pH 7.3) containing 50 mM 2-(N-morpholino)ethanesulfonic acid, 50 mM trisaminomethane, 0.1% sodium dodecylsulfate (*w/v*), and 20 mM ethylenediaminetetraacetic acid. Proteins were transferred to a nitrocellulose membrane and probed with antibodies against Rieske FeS protein (Agrisera cat. no. AS08330, Vännäs, Sweden) at 1:3000 dilution, and Rubisco small subunit protein (Agrisera cat. no. AS07259, Vännäs, Sweden) at 1:10,000 dilution. Quantification of Western blots was performed with Image Lab software (Biorad Laboratories, Hercules, CA, USA).

Leaf chlorophyll was extracted in 80% acetone and sample absorbance was determined with an Agilent Cary 60 UV-Vis spectrophotometer (Agilent Technologies, USA) at 663.2 nm and 646.8 nm against an 80% acetone blank. Chlorophyll A content was determined from sample absorbance according to Lichtenthaler [46].

### 4.5. Measurement of Total Nitrogen and Carbon

Additional leaf discs from where photosynthetic measurements were conducted were used for measurements of leaf carbon (C) and nitrogen (N) content. Leaf discs (0.353 cm^2^) were oven-dried at 80 °C, weighed, and then ground to powder in a ball mill. Percentage of N and C was determined on the ground tissues using an Isoprime isotope-ratio mass spectrometer (Elementar).

### 4.6. Statistical Analyses

All statistical analyses were completed in GraphPad Prism^TM^ 9.2.0 (GraphPad Software, San Diego, CA, USA). Where two-way ANOVA results were significant, a Tukey’s multiple comparisons test was used to identify significant pair-wise differences arising from CO_2_ treatment or plant genotype. *p*-values < 0.05 were considered significant.

## Figures and Tables

**Figure 1 plants-12-00441-f001:**
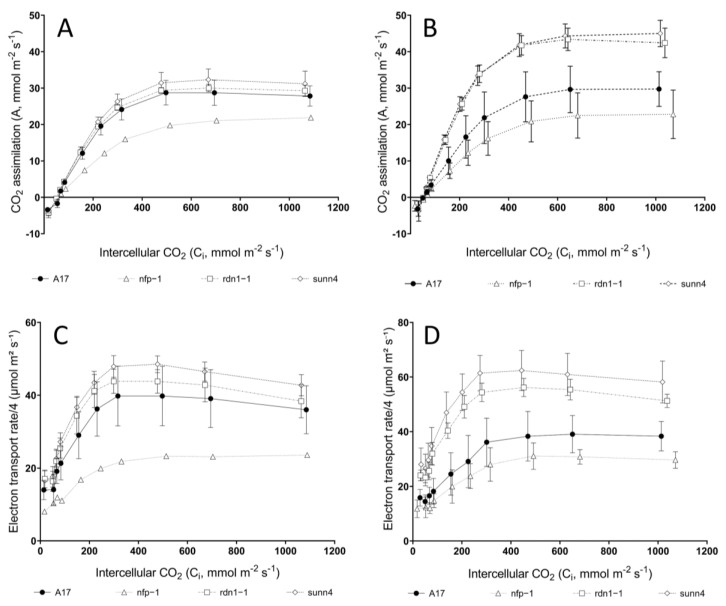
CO_2_ responses curves (**A**,**B**) and electron transport rates (**C**,**D**) of *Medicago truncatula* nodulation mutants (open symbols) were comparable to wild-type plants (solid symbol) when grown under ambient CO_2_ levels (**A**,**C**), but differed significantly under elevated CO_2_ levels (**B**,**D**). Measurements were taken at 25 °C and 21% O_2_. Error bars represent standard deviation; see footnotes of Table 1 for sample sizes.

**Figure 2 plants-12-00441-f002:**
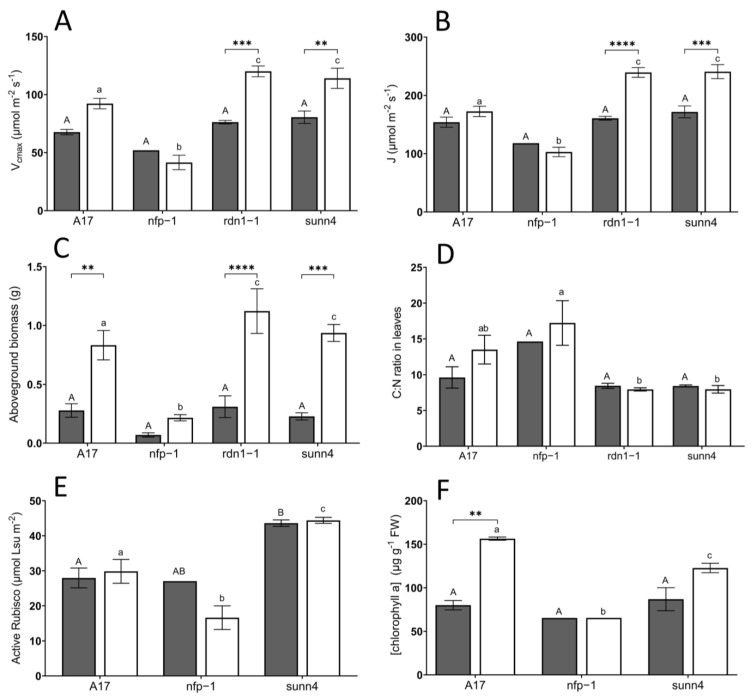
Comparisons of (**A**) maximum Rubisco activity, V_cmax_, (**B**) maximum electron transport, J, (**C**) above-ground biomass, (**D**) carbon/nitrogen ratio, (**E**) active Rubisco content, and (**F**) chlorophyll A concentration in leaves of *Medicago truncatula* lines grown at ambient (aCO_2_, dark grey bars) and elevated (eCO_2_, white bars) conditions. Error bars represent standard error; see footnotes of Table 1 for sample sizes. Asterisks indicate level of statistical significance (** = *p* < 0.01, *** = *p* < 0.001, **** = *p* < 0.0001) in pairwise differences between CO_2_ treatments. Uppercase letters describe significant groupings at ambient CO_2_ levels while lowercase letters describe significant groups at elevated CO_2_ levels; values which do not share a common letter display statistical significance from each other.

**Figure 3 plants-12-00441-f003:**
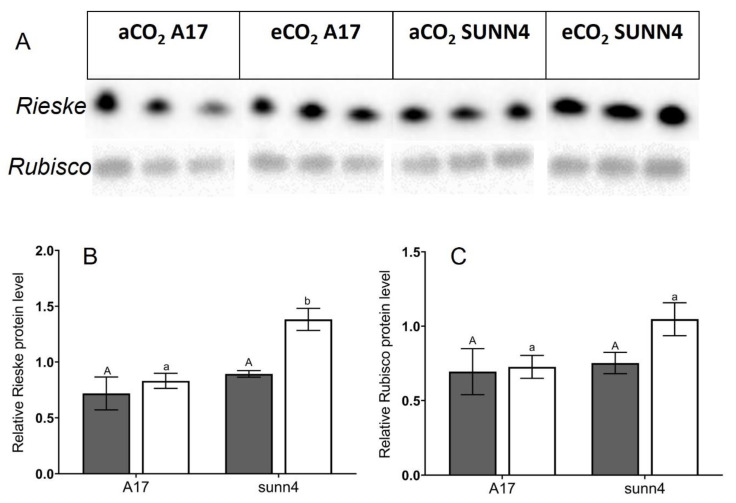
(**A**) Western Blot analysis of Rieske and Rubisco protein content in wild-type and *sunn-4* super-nodulating *Medicago truncatula* lines grown at ambient (aCO_2_, dark grey bars) and elevated (eCO_2_, white bars) conditions. (**B**) *Medicago truncatula sunn4* super-nodulating mutant displays significantly higher relative Rieske protein levels than wild-type and non-nodulating plants under elevated CO_2_ levels (white bars), but not under ambient CO_2_ levels (dark grey bars). (**C**) The relative Rubisco protein level of the *Medicago truncatula sunn4* super-nodulating mutant do not differ significantly from wild-type plants under both ambient (dark grey bars) and elevated CO_2_ levels (white bars). Error bars represent standard error; see footnotes of Table 1 for sample sizes. Uppercase letters describe significant groupings at ambient CO_2_ levels while lowercase letters describe significant groups at elevated CO_2_ levels; values which do not share a common letter display statistical significance from each other.

**Figure 4 plants-12-00441-f004:**
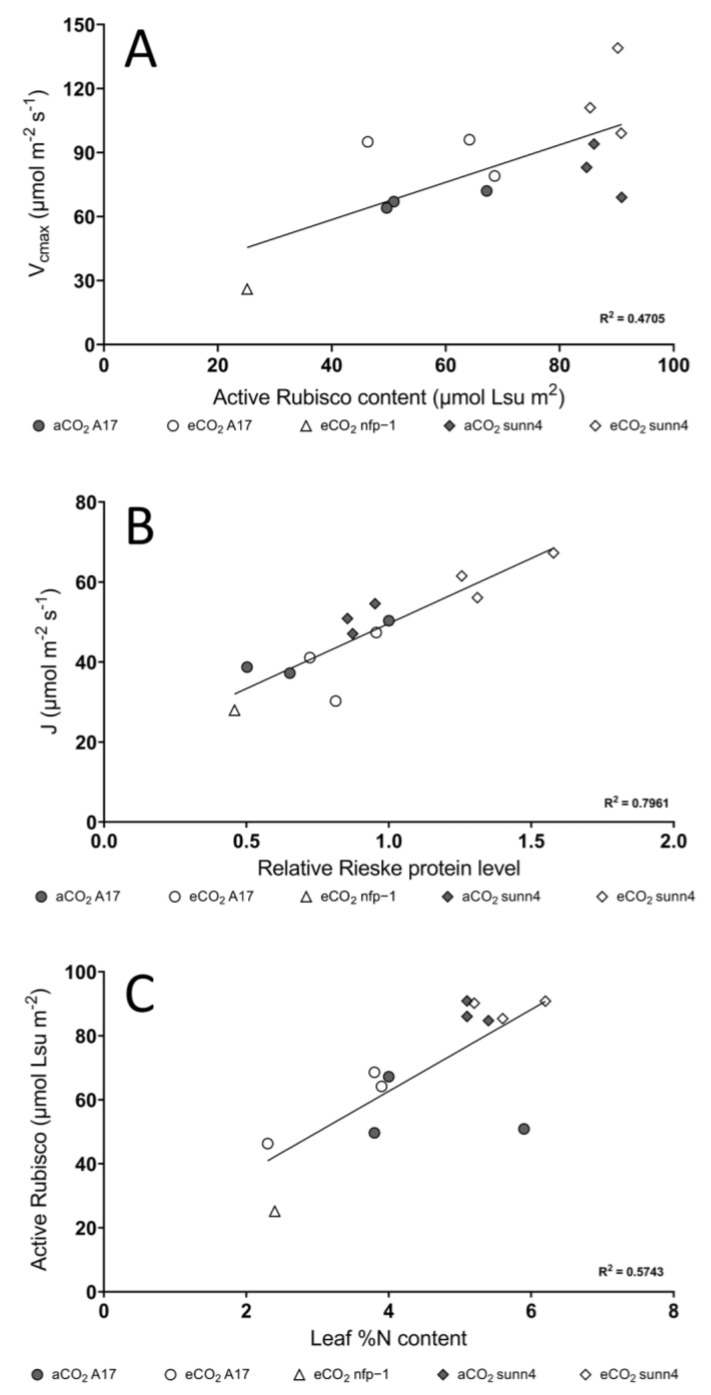
The relationship between (**A**) active Rubisco content and V_cmax_, (**B**) relative Rieske protein level and (J), (**C**) leaf %N content and active Rubisco content in *Medicago truncatula* wild-type plants (A17, circles), a super-nodulating mutant *sunn4* (diamonds) and the non-nodulating mutant *nfp-1* (triangles) at ambient CO_2_ (aCO_2_, dark markers) and elevated CO_2_ (eCO_2_, white markers).

**Table 1 plants-12-00441-t001:** Various phenotypic traits in wild-type (A17), super-nodulating (*rdn1-1* and *sunn4*) and non-nodulating (*nfp1*) *Medicago truncatula* plants grown at ambient (aCO_2_, dark grey cells) and elevated (eCO_2_, white cells) conditions. Uncertainty (±) is derived from standard error; cells with a dash (-) were not quantified for the relevant trait. Values in bold indicate statistically significant (*p* < 0.05) pairwise differences between CO_2_ treatments. Uppercase letters describe significant groupings at ambient CO_2_ levels while lowercase letters describe significant groups at elevated CO_2_ levels; values within a row that do not share a common letter display statistical significance (*p* < 0.05) from each other. * *n* = 1 for aCO_2_ *nfp1*; *n* = 3 for aCO_2_ A17, *n* = 4 for eCO_2_ A17, eCO_2_ *nfp1*, eCO_2_ *sunn4*, aCO_2_ *rdn1-1*, aCO_2_ *sunn4*; *n* = 5 for eCO_2_ *rdn1-1*. # *n* = 1 for aCO_2_ *nfp1*, eCO_2_ *nfp1*; *n* = 3 for aCO_2_ A17, aCO_2_ *sunn4*, eCO_2_ A17, eCO_2_ *sunn4*. † *n* = 5 for all treatments. ‡ *n* = 1 for aCO_2_ *nfp1*, eCO_2_ *nfp1*; *n* = 3 for aCO_2_ A17, aCO_2_ *sunn4*, eCO_2_ A17, eCO_2_ *sunn4*.

	Genotype and Treatment							
	aCO_2_ A17	eCO_2_ A17	aCO_2_ *nfp1*	eCO_2_ *nfp1*	aCO_2_ *rdn1-1*	eCO_2_ *rdn1-1*	aCO_2_ *sunn4*	eCO_2_ *sunn4*
V_cmax_ (µmol m^−2^ s^−1^) *	68 ± 2.6 ^A^	92 ± 4.4 ^a^	52 ^A^	41 ± 6.1 ^b^	76 ± 1.4 ^A^	**120 ± 4.8 ^c^**	81 ± 5.3 ^A^	**114 ± 8.6 ^c^**
J (µmol m^−2^ s^−1^) *	154 ± 8.8 ^A^	173 ± 8.9 ^a^	118 ^A^	103 ± 8.0 ^b^	161 ± 3.2 ^A^	**239 ± 12 ^c^**	172 ± 10 ^A^	**154 ± 8.8 ^c^**
Stomatal conductance at 400 ppm (mol m^−2^ s^−1^) *	0.52 ± 0.02 ^A^	0.41 ± 0.06 ^a^	0.38 ^A^	0.32 ± 0.05 ^a^	0.45 ± 0.02 ^A^	0.51 ± 0.08 ^a^	0.47 ± 0.04 ^A^	0.44 ± 0.04 ^a^
Electron transport rate/4 at 400 ppm (µmol m^−2^ s^−1^) *	42 ± 4.1 ^A^	37 ± 4.2 ^a^	23 ^AB^	29 ± 4.3 ^a^	48 ± 2.4 ^AC^	59 ± 1.8 ^b^	51 ± 1.5 ^AC^	64 ± 3.0 ^b^
Active Rubisco content (µmol LSU m^−2^) ^#^	28 ± 2.8 ^A^	30 ± 3.4 ^a^	27 ^AB^	17 ± 3.4 ^b^	-	-	44 ± 0.94 ^B^	44 ± 0.86 ^c^
[Chlorophyll A] (µg g^−1^ FW) ^#^	80 ± 5.3 ^A^	**156 ± 1.8 ^a^**	63 ^A^	65 ^b^	-	-	87 ± 13 ^A^	123 ± 5.4 ^c^
Specific leaf area (m^−2^ g) *	0.035 ± 0.003 ^A^	0.021 ± 0.003 ^a^	0.027 ^A^	0.025 ± 0.003 ^a^	0.026 ± 0.004 ^A^	0.029 ± 0.002 ^a^	0.030 ± 0.002 ^A^	0.023 ± 0.004 ^a^
Above-ground biomass (g) ^†^	0.28 ± 0.058 ^A^	**0.83 ± 0.125 ^a^**	0.070 ± 0.018 ^A^	0.22 ± 0.027 ^b^	0.31 ± 0.093 ^A^	**1.12 ± 0.19 ^c^**	0.23 ± 0.031 ^A^	**0.94 ± 0.072 ^c^**
Relative Rieske protein content ^‡^	0.72 ± 0.15 ^A^	0.83 ± 0.068 ^a^	-	0.46 ^a^	-	-	0.89 ± 0.03 ^A^	1.38 ± 0.10 ^b^
Relative Rubisco protein content ^‡^	0.70 ± 0.16 ^A^	0.73 ± 0.077 ^a^	-	0.48 ^a^	-	-	0.75 ± 0.071 ^A^	1.05 ± 0.11 ^a^
%N in leaves *	4.57 ± 0.67 ^A^	3.05 ± 0.46 ^a^	2.80 ^A^	2.34 ± 0.44 ^a^	5.18 ± 0.21 ^A^	5.36 ± 0.15 ^b^	5.23 ± 0.08 ^A^	5.38 ± 0.36 ^b^
%C in leaves *	43.9 ± 1.3 ^A^	**41.1 ± 0.3 ^ab^**	41.0 ^A^	40.3 ± 0.5 ^a^	43.7 ± 0.3 ^A^	42.7 ± 0.5 ^b^	44.1 ± 0.4 ^A^	42.9 ± 0.4 ^b^
C:N ratio in leaves *	9.61 ± 1.48 ^A^	13.5 ± 2.0 ^ab^	14.6 ^A^	17.2 ± 3.1 ^a^	8.45 ± 0.36 ^A^	7.95 ± 0.22 ^b^	8.43 ± 0.14 ^A^	7.96 ± 0.53 ^b^

## Data Availability

All raw data for this study is available online as supplementary data and includes uncropped western blots, raw data for CO_2_ response curves, CABP assays, chlorophyll content and C and N analysis.

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
