# Peer review of "Photosynthetic Gains in Super-Nodulating Mutants of *Medicago truncatula* under Elevated Atmospheric CO_2_ Conditions"

_plants, 2023, doi:10.3390/plants12030441_

Round 1

Reviewer 1 Report

This publication presents interesting results related to the photosynthetic and physiological traits of super-nodulating Medicago truncatula mutants grown under elevated atmospheric CO2 conditions. The adopted approach in this study was consistent since it could provide more insights into understanding carbon and nitrogen partitioning under high CO2 concentrations and promoting crop production in changing environment.

The manuscript was well introduced, and the authors adopted convincing methods with a discussion of the different obtained results. However, the manuscript needs minor revisions to be suitable for publication in Plants.

General comment

Comment 1: The English of this manuscript needs minor improvements.

Comment 2: The section titles should be more specific

Comment 3: please add line numbers to make it easier to provide comments

Other comments

- Abstract

Keywords: please add “super-nodulating mutants” and “Medicago truncatula” as keywords.

- Introduction

- Results:

Figure 2: Please mention  aCO2 and eCO2 conditions to differentiate gray and white bars. The same for Figure 3.

Section 2.2. title: please make it more specific.

Please place Figure 4 in the Results section.

Please add a paragraph to describe Figure 4 results.

- Discussion:

The fourth line:  please change “We found biomass” to “We found that biomass”

- M&M

The section title: please change “Methods” to “Material and Methods”.

The second paragraph: Please add a space between the value and the unit. Please check throughout the manuscript.

The last paragraph of 4.1. subsection: please provide the significance of “OD” at the first appearance in the text. Please check throughout the manuscript.

The first paragraph of 4.4. subsection: please change “ Ruuska et al. (Ruuska et al., 1998)” to “Ruuska et al. (1998)”.

Reviewer 2 Report

The author presented a study that AON mutants super-nodulate perform better at eCO2 through improved photosynthesis, which may be attributed to optimal carbon and nitrogen allocation within the mutants. The results further conform the previously published results that legumes experience greater stimulation of photosynthesis under eCO2. However, there are some points needed to be addressed from my points.

Abstract:

The key points were highlighted.

Introduction:

This section is fine and highlight objectives of this study.

Results:

Figure: change “NFP” “RDN” “SUNN4”to “nfp1” “rnd1-1” “sunn4”.

Please check the order of the pictures in Figure 2, the section of results does not correspond to the position of the pictures.

Figure3:

3(A) Please explain “four different Medicago truncatula lines”, I just find two lines in the picture.

3(B) I don't find “non-nodulating plants” in the picture, and there are only results of eCO2 nfp1 and not aCO2 nfp1 in Table 1.

3(C) I don't find “nfp1 non-nodulating” in the picture, and there are only results of eCO2 nfp1 and not aCO2 nfp1 in Table 1.

Discussion:

1Figure4: Please add legends to illustrate the meanings of the different shapes.

2What is the underlying mechanism or significance of the positive affiliation with increased leaf %N?

3In western experiments, rnd1-1(derived from Jemalong A17), which is consistent with the wild-type(Jemalong A17) background, can more accurately explain the results than sunn4 (Jemalong J5), There may be some differences between two genotypes that we can't be sure about. Although rdn1-1 and sunn4 mutants performed comparably within all the phenotypic traits assessed in this study under ambient and elevated CO2 conditions, contrary results were found in previous studies.

4Please standardize the full names and abbreviations.

M&M:

Pleases delete “470nm”.

Pleases rephase the section of “4.6 Statistical analyses” by science writing.

Round 2

Reviewer 2 Report

This version can be accepted.